# Predictive Control of Modular Multilevel Converters: Adaptive Hybrid Framework for Circulating Current and Capacitor Voltage Fluctuation Suppression †

**Junda Li, Zhenbin Zhang \*, Zhen Li and Oluleke Babayomi**

School of Electrical Engineering, Shandong University, Jinan 250061, China; 202114559@mail.sdu.edu.cn (J.L.); zhenli0901@sdu.edu.cn (Z.L.); oluleke.babayomi@mail.sdu.edu.cn (O.B.)

\* Correspondence: zbz@sdu.edu.cn; Tel.: +86-0531-8839-2002

† This paper is an extended version of our paper published in 2023 IEEE International Conference on Predictive Control of Electrical Drives and Power Electronics (PRECEDE), Wuhan, Hubei province, China, 16–19 June 2023; pp. 1–6.

**Abstract:** Modular multilevel converters (MMCs) are widely used in voltage-sourced, converter-based high-voltage DC systems due to their modular design, scalability, and fault tolerance capabilities. In MMCs, multi-variable control objectives can be employed by using model predictive control (MPC) due to its fast dynamic response and ease of implementation. Nonetheless, conventional MPC techniques for MMCs have shortcomings, including high computational requirements, poor circulating current, and capacitor voltage fluctuation suppression. First, this study proposes an adaptive MPC technique that adapts the number of candidate combinations to the steady and transient states, significantly reducing the computational burden. Second, an improved hybrid combination of an MPC with a proportional-resonance (PR) controller enhances the circulating current and capacitor voltage fluctuation suppression performance. According to the phase difference between the circulating current and the capacitor voltage, the circulating current and capacitor voltage can be suppressed at different times by changing the circulating current reference of the PR controller. The switching frequency can be reduced by using the PR controller's output to adjust the input submodule number instead of changing the duty cycle. The proposed techniques were validated by simulations and experimental case studies with a three-phase grid-connected MMC.

**Keywords:** modular multilevel converters; model predictive control; capacitor voltage fluctuation; circulating current; fast optimization



## 1. Introduction

Modular multilevel converters (MMCs) are widely used in voltage-sourced, converter-based high-voltage DC (VSC-HVDC) systems due to their modular design, easy scaling, and fault tolerance capabilities [1]. These features also make MMCs suitable for offshore wind power integration [2]. The increased installed capacity of offshore wind power systems has led to a greater demand for efficient power allocation and suppression of voltage and power fluctuations [3]. These requirements can also be met by MMCs.

MMCs need to fulfill the following control objectives: (1) current and power tracking, (2) circulating current suppression, and (3) capacitor voltage fluctuation suppression. Traditional linear controllers, such as proportional–integral (PI) or proportional-resonance (PR) controllers, have difficulty effectively achieving multiple nonlinear control objectives due to their cascading structures, which can lead to bandwidth reduction in multi-objective control. Model predictive control (MPC) can deal with multiple control objectives while ensuring good dynamic performance and can effectively deal with nonlinear constraints [4]. Therefore, MPC has become an excellent choice for controlling high-power, complex, multi-level converters [5,6].

A direct MPC for MMCs, which incorporates the three control objectives into one cost function with weighting factors, was proposed originally in [7]. However, direct MPC requires traversing all switch states. The more submodules (SMs) there are, the heavier the computational burden is exponentially. Many methods based on weighting factors [8,9] and cascade optimization modes [10,11] have been proposed to solve this issue. Ref. [12] proposed a sequential-optimization MPC using the SM input number. The switching state is replaced by the SM input number, which linearly increases with the number of SMs on the bridge arm. If the MMC has $N$ SMs on each arm, it only requires $N + 1$ calculations per phase. However, the amount of computation is still too high. The work in [13,14] adopted a simplified-calculation MPC. It just considers three SMs for the insertion number combinations, which is close to the optimal insertion number obtained at the last interval, and only calculates three cost functions. Due to this method of only calculating three candidate combinations each time, it requires more time to reach the steady state when the reference has a sudden change. Therefore, it has poor dynamic performance. Finding an MPC that balances computational burden and dynamic performance is necessary.

Furthermore, MMCs have the limitation of inner circulating currents being at twice the fundamental frequency, leading to an increase in the system's heating capacity [15]. When using the aforementioned MPC methods, which rely on weighting factors or sequential optimization modes to achieve multi-objective control, the subordinate control of the circulating current can lead to poor suppression performance. Therefore, it is essential to design a control strategy to suppress circulating current under the MPC framework. Ref. [16] used a PI controller to suppress circulating current, but its performance was poor. The authors of [17] proposed a hybrid MPC framework that combines MPC and PR controllers. The output of the PR controllers is used to adjust the switching signals generated by the current control, considerably suppressing the circulating current. However, it generates multiple switching actions during one single sampling interval, increasing the switching frequency.

Additionally, SM capacitors are the largest and most expensive components in MMCs [18]. Reducing the capacitor voltage fluctuation can reduce the capacitor capacity required for the SM, which has important practical engineering significance [19]. Ref. [20] proposed a direct control method based on the recent level modulation. This method reduces the circulating current to zero, but improper control can increase the amplitude of capacitor voltage changes [21]. Ref. [22] proposed to focus on capacitor voltage fluctuations under all operating conditions but relied on offline table lookup. Wang et al. [23] introduced real-time output current to generate circulating current reference values online through different methods, but the method required high current measurement accuracy. The authors of [24] indirectly generated a circulating current reference value by controlling the capacitor voltages in the *dq* coordinate system, but the controller needed to decouple between phases. Ref. [25] proposed a third-harmonic injection method that reduced capacitor voltage fluctuations but impacted the AC side voltage of the inverter. Ref. [26] proposed a trajectory planning-based third-harmonic voltage injection method. Dong et al. [27] proposed a coupling injection strategy for the third-harmonic voltage and second-harmonic current, which improved the modulation ratio by injecting third-harmonic voltage and achieved better control performance in combination with the second-harmonic current injection strategy. However, the two harmonic injection schemes have strong coupling, and the mechanism still needs further exploration.

This study was motivated by the research gaps regarding high-dynamic-performance multi-objective optimization of MMC capacitor voltage balancing with circulating current suppression while ensuring low computational burden and reduced switching losses. Therefore, we propose an improved adaptive hybrid predictive control framework for MMCs. The main contributions are as follows:

1.　To solve the existing issue of the high computational requirement in [13], this work proposes an adaptive MPC method. It dynamically adjusts the number of candidate combinations by distinguishing the operation modes between steady and transient

states, significantly reducing the computational burden while maintaining the fast dynamics during transient states;

2.  To balance capacitor voltage and circulating current, this work developed an improved hybrid control framework, which uses an MPC to control the current and two PR controllers to control the circulating current and capacitor voltage. Unlike the method in [17], the output of the PR controller is used to adjust the number of inserted SMs instead of modifying the switching signals, which maintains the single-interval, single-switching characteristics of the MPC and hence reduces the total switching frequency;

3.  A simulation and experimental results verified the excellent control performance of the proposed method;

The contents of this article are organized as follows. In Section 2, the system models of the grid-tied MMC are presented. Section 3 reviews the classical simplified optimization hybrid predictive control framework, and in Section 4, we introduce the proposed adaptive hybrid predictive control framework. Section 5 presents the verification and analysis of the proposed method. Finally, Section 6 concludes this paper.

## 2. System Model of Grid-Connected MMC

The continuous and discrete time models of the grid-connected MMC are shown in Figure 1, which will be used in subsequent sections. Each $x$ phase ($x$ = a, b, c) includes the upper bridge arm (represented by the subscript p) and the lower bridge arm (represented by the subscript n). Each bridge arm comprises $N$ SMs and one bridge arm inductance. Each SM has two IGBTs and one SM capacitor. The capacitor voltage of an SM is $V_C$.

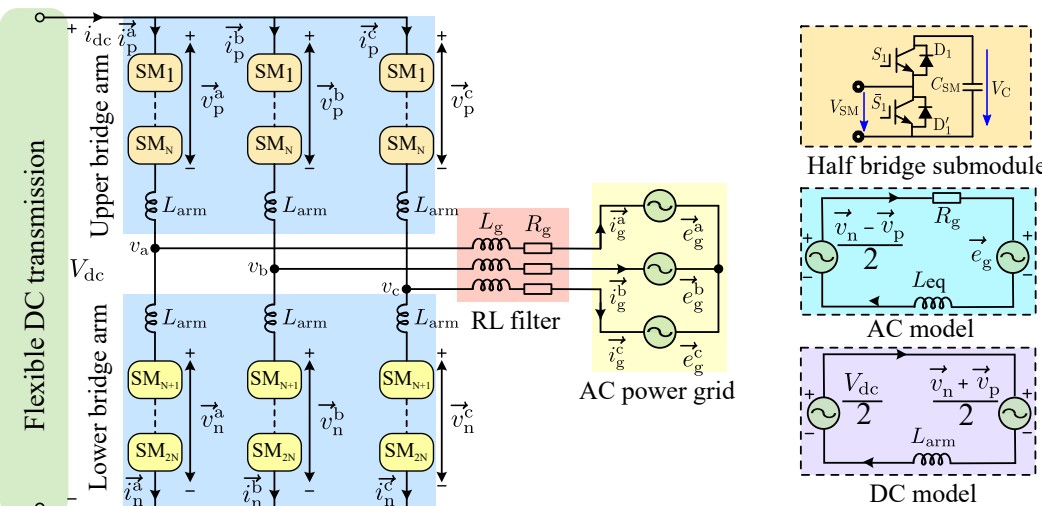

**Figure 1.** Topology of grid-tied modular multilevel converter (MMC).

If the capacitor voltage of all SMs $V_C$ in a bridge arm is equal, the AC voltage of the bridge arm ($\vec{v}_p^x = N_p^x V_C$ and $\vec{v}_n^x = N_n^x V_C$) is proportional to the number of SMs working on the bridge arm ($N_p^x$ and $N_n^x$) [28]. Additionally, the AC outlet side of the MMC transmits energy to the grid through a resistor inductance filter. $L_g$ and $R_g$ are the filtering inductance and filtering resistance of the resistor inductance filter, respectively, while $\vec{i}_g^x$ and $\vec{e}_g^x$ are the AC grid current and AC voltage. The output grid voltage on the AC side of the MMC is as follows:

$$\vec{v}_o^x = \frac{\vec{v}_n^x - \vec{v}_p^x}{2} = \frac{1}{2}\left(\sum_{i=0}^{N} \vec{V}_{Cn}^{xi} S_n^{xi} - \sum_{i=0}^{N} \vec{V}_{Cp}^{xi} S_p^{xi}\right) = \frac{(N_n^x - N_p^x)V_C}{2} \tag{1}$$

It can be seen that the AC voltage of the MMC depends on the number of SMs working on the upper and lower bridge arms. The continuous time model of the output AC and DC MMC circuits is as follows:

$$L_{eq}\frac{d\vec{i}_g^x}{dt} + R_g\vec{i}_g^x = \frac{\vec{v}_n^x - \vec{v}_p^x}{2} - \vec{e}_g^x, \quad L_{arm}\frac{d\vec{i}_z^x}{dt} + \frac{\vec{v}_p^x + \vec{v}_n^x}{2} = \frac{V_{dc}}{2} \tag{2}$$

where $L_{eq} = L_{arm}/2 + L_g$ is the equivalent inductance of the MMC. $\vec{i}_z^x = i_{dc}/3 + i_{cir}$ is the circulating current generated by the x phase. $\vec{i}_z^x$ is a DC circulating current that transfers energy to the outside of the MMC without generating heat. $i_{cir}$ is the internal flow of the AC circulating current that only consumes energy. The continuous time model between the capacitor voltage and bridge arm current of the x phase SM is as follows:

$$C_{SM}\frac{d\vec{V}_{Cp}^{xi}}{dt} = S_p^{xi}\vec{i}_p^x, \quad C_{SM}\frac{d\vec{V}_{Cn}^{xi}}{dt} = S_n^{xi}\vec{i}_n^x \tag{3}$$

where $C_{SM}$ is the SM capacitance, $\vec{V}_{Cp}^{xi}$ and $\vec{V}_{Cn}^{xi}$ are the capacitor voltages of the *i*-th SM on the *x* phase's two bridge arms, and $S_p^{xi}$ and $S_n^{xi}$ are the *i*-th SM's switching states. Combining Equations (2) and (3), the discrete time models for the grid current and circulating current are as follows:

$$\vec{i}_g^x[k+1] = \Phi_o\vec{i}_g^x[k] + \Gamma_o\left((N_n^x\vec{V}_{Cn}^x - N_p^x\vec{V}_{Cp}^x) - 2\vec{e}_g^x[k]\right) \tag{4}$$

$$\vec{i}_z^x[k+1] = \vec{i}_z^x[k] + \Lambda_o\left(V_{dc} - (N_n^x\vec{V}_{Cn}^x + N_p^x\vec{V}_{Cp}^x)\right) \tag{5}$$

where *k* is the *k*-th control cycle. $\Phi_o = 1 - T_s R_g/L_{eq}$, $\Gamma_o = T_s/(2L_{eq})$, $\Lambda_o = T_s/(2L_{arm})$. $\vec{V}_{Cp}^x$ and $\vec{V}_{Cn}^x$ are the average values of *x* phase capacitor voltages.

## 3. Classical Simplified Optimization Hybrid Predictive Control Framework

This section introduces the classical sequential optimization MPC method based on the voltage level [12] and the simplified optimization MPC [13]. These methods based on single multi-objective cost functions lead to the coupling of multiple control objectives in the MMC. This type of method can only prioritize the control performance of the grid current, resulting in subordinate control of the capacitor voltage and circulating current. Then, we introduce the hybrid MPC with a PR controller proposed by [17]. It uses a hybrid MPC framework with a quasi-PR controller. The quasi-PR controller controlled in parallel increases the switching actions related to the capacitor voltage and circulating current, significantly suppressing the circulating current and capacitor voltage fluctuation.

### 3.1. Classical Simplified Optimization MPC

Figure 2 presents the classical sequential optimization MPC method based on voltage level. It has a clear priority without weighting factors. It uses multiple cost functions in a sequential mode, and each contains only one control objective. First, the active power reference $P^*$ is obtained by the outer DC voltage controller. The three-phase grid voltage at the time interval $k + 1$ can be calculated by:

$$\vec{e}_g^x[k+1] = 3(\vec{e}_g^x[k] - \vec{e}_g^x[k-1]) + \vec{e}_g^x[k-2] \tag{6}$$

The grid voltages in the $\alpha\beta$ coordinate system $\vec{e}_g^\alpha[k+1]$ and $\vec{e}_g^\beta[k+1]$ are calculated. According to power references $P^*$ and $Q^*$, the current references ($\vec{i}_g^\alpha[k+1]$ and $\vec{i}_g^\beta[k+1]$) in the $\alpha\beta$ coordinate system are as follows:

$$\vec{i}_g^{\alpha*}[k+1] = \frac{\vec{e}_g^\alpha[k+1]P^*[k] + \vec{e}_g^\beta[k+1]Q^*[k]}{\vec{e}_g^\alpha[k+1]^2 + \vec{e}_g^\beta[k+1]^2}, \vec{i}_g^{\beta*}[k+1] = \frac{\vec{e}_g^\beta[k+1]P^*[k] - \vec{e}_g^\alpha[k+1]Q^*[k]}{\vec{e}_g^\alpha[k+1]^2 + \vec{e}_g^\beta[k+1]^2} \tag{7}$$

The three-phase current reference $\vec{i}_g^{x*}[k+1]$ is calculated with the inverse Clarke transformation. The first control objective is the grid current, and its cost function $J_i^x$ is as follows:

$$J_i^x(N_p^x[k+1], N_n^x[k+1]) = (\vec{i}_g^x[k+2] - \vec{i}_g^{x*}[k+1])^2 \tag{8}$$

In the simplified optimization MPC [13], when calculating the predicted current values using Equation (4), instead of considering all possible combinations, only the insertion number combinations near the optimal combination obtained from the previous control interval $(N_p^x[k], N_n^x[k])$ are selected as the candidate combinations. These combinations are $(N_p^x[k], N_n^x[k])$, $(N_p^x[k]+1, N_n^x[k]-1)$, and $(N_p^x[k]-1, N_n^x[k]+1)$. After minimizing the cost function of Equation (8), the optimal SM insertion number combinations $(N_p^{x1}[k+1]$ and $N_n^{x1}[k+1])$ are obtained.

By adding $+1$, $-1$, and $+0$ to the optimal combination from the current control, three new candidate combinations can be obtained $((N_p^{x1}[k+1], N_n^{x1}[k+1])$, $(N_p^{x1}[k+1]+1,$ $N_n^{x1}[k+1]+1)$, and $(N_p^{x1}[k+1]-1, N_n^{x1}[k+1]-1))$ for the circulating current control. The cost function of the circulating current $J_z^x$ is as follows:

$$J_z^x(N_p^{x\text{opt}}[k+1], N_n^{x\text{opt}}[k+1]) = (\vec{i}_z^x[k+2] - \vec{i}_z^{x*}[k+1])^2 \tag{9}$$

After calculating the predicted circulating current value using the three candidate combinations, the optimal input number combinations $N_p^{x\text{opt}}[k+1]$ and $N_n^{x\text{opt}}[k+1]$ are selected by selecting the candidate combinations with the minimum cost function. As shown in Figure 3, the simplified optimization MPC only needs to calculate six cost functions per cycle, which does not increase with the growth in $N$. However, when the system undergoes drastic changes, the simplified optimization MPC may require more control cycles for the calculation. It can only move one step per cycle, gradually transitioning to the next ideal combination, resulting in a slower dynamic response.

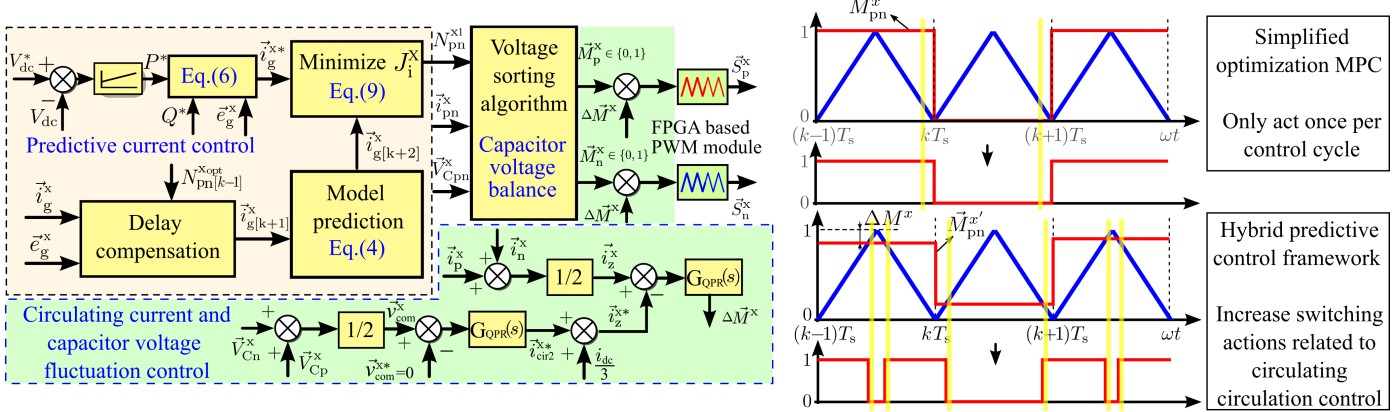

**Figure 2.** The simplified optimization hybrid predictive control framework.

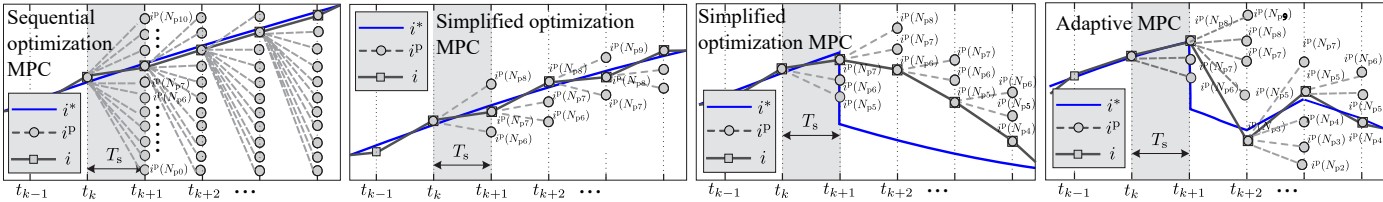

**Figure 3.** Control diagrams for sequential optimization MPC, simplified optimization MPC and proposed adaptive MPC.

### 3.2. Classical Quasi-PR Circulating Current and Capacitor Voltage Fluctuation Suppression Controller

The method based on a single multi-objective cost function or multiple multi-objective cost functions leads to the coupling of multiple control objectives in the MMC. This type of method can only prioritize the control performance of the grid current, resulting in subordinate control of the circulating current. To improve the circulating current performance of the MPC, Ref. [17] proposed a hybrid MPC with a PR controller, where the MPC is used to manage power and current and the quasi-PR controller is used to suppress circulating current.

Strong coupling exists when controlling capacitor voltage fluctuation through the second-harmonic current injection and the third-harmonic voltage injection. To balance capacitor voltage and circulating current, a capacitor voltage fluctuation suppression outer loop controller and a circulating current suppression inner loop controller were used to suppress the second-harmonic current and harmonic voltage injection. The capacitor voltage fluctuation controller was based on the suppression of the circulating current.

The transfer function of the quasi-PR controller for the MMC is as follows:

$$G_{\text{QPR}}(s) = K_{\text{p}} + \frac{2K_{\text{r}}\omega_c s}{s^2 + 2\omega_c s + \omega_o{}^2} \tag{10}$$

where $K_{\text{p}}$, $K_{\text{r}}$, $\omega_{\text{c}}$, and $\omega_{\text{o}}$ are the proportional gain, resonant gain, cutoff frequency, and resonant frequency of the quasi-PR controller; based on this, a PR-based circulating current controller can be designed. First, $\vec{i}_z^{\text{x}} = (\vec{i}_p^{\text{x}} + \vec{i}_n^{\text{x}})/2$ is used to calculate the three-phase circulating current value of the grid-tied MMC, and it is compared with the circulating current reference value $\vec{i}_z^{\text{x}*}[k+1]$. Then, the compared value is sent to the quasi-PR controller, and the resonance frequency of the quasi-PR controller $\omega_{\text{o}}$ is calculated and set to twice the fundamental frequency $2\omega_{\text{g}}$. It is used to amplify the component of the circulating current with twice the fundamental frequency. According to [29], the capacitor voltage is as follows:

$$\vec{V}_{\text{Cp}}^{\text{x}} = \frac{V_{\text{dc}}}{N} - \frac{\vec{i}_{\text{g}}^{\text{x}}}{4\omega_{\text{g}}C_{\text{SM}}}\cos(2\omega_{\text{g}}t + \varphi) + \frac{mi_{\text{dc}}}{6\omega_{\text{g}}C_{\text{SM}}}\cos(\omega_{\text{g}}t) - \frac{mi_{\text{cir2}}}{4\omega_{\text{g}}C_{\text{SM}}}\sin(\omega_{\text{g}}t + \theta_2)$$
$$- \frac{i_{\text{cir2}}}{4\omega_{\text{g}}C_{\text{SM}}}\cos(2\omega_{\text{g}}t + \theta_2) + \frac{m\vec{i}_{\text{g}}^{\text{x}}}{16\omega_{\text{g}}C_{\text{SM}}}\cos(2\omega_{\text{g}}t + \varphi) + \frac{mi_{\text{cir2}}}{12\omega_{\text{g}}C_{\text{SM}}}\sin(3\omega_{\text{g}}t + \theta_2) \tag{11}$$

$$\vec{V}_{\text{Cn}}^{\text{x}} = \frac{V_{\text{dc}}}{N} + \frac{\vec{i}_{\text{g}}^{\text{x}}}{4\omega_{\text{g}}C_{\text{SM}}}\cos(2\omega_{\text{g}}t + \varphi) - \frac{mi_{\text{dc}}}{6\omega_{\text{g}}C_{\text{SM}}}\cos(\omega_{\text{g}}t) + \frac{mi_{\text{cir2}}}{4\omega_{\text{g}}C_{\text{SM}}}\sin(\omega_{\text{g}}t + \theta_2)$$
$$- \frac{i_{\text{cir2}}}{4\omega_{\text{g}}C_{\text{SM}}}\cos(2\omega_{\text{g}}t + \theta_2) + \frac{m\vec{i}_{\text{g}}^{\text{x}}}{16\omega_{\text{g}}C_{\text{SM}}}\cos(2\omega_{\text{g}}t + \varphi) - \frac{mi_{\text{cir2}}}{12\omega_{\text{g}}C_{\text{SM}}}\sin(3\omega_{\text{g}}t + \theta_2) \tag{12}$$

where $m = 2\vec{v}_{\text{g}}^{\text{x}}/V_{\text{dc}}$, $i_{\text{cir2}}$ is the secondary circulating current, and $\theta_2$ and $\varphi$ are the phase angles between the secondary circulating current and grid current. The classical circulating current suppression method suppresses the amplitude of the secondary circulating current component to 0, eliminating the third component of the capacitor voltage and reducing the amplitude of some fundamental and secondary components. However, the secondary components with a high proportion of harmonic components cannot be eliminated. If the amplitude and phase angle of the secondary circulating current can be controlled to reach a specific value, then the second-harmonic component of the capacitor voltage can be completely eliminated [29]. The predicted circulating current values are as follows:

$$i_{\text{cir2}} = m\vec{i}_{\text{g}}^{\text{x}}\sin(2\omega_{\text{g}}t + \varphi + \pi/2)/4 \tag{13}$$

When the secondary circulating current is at the predicted value, the second-harmonic component of the capacitor voltage is eliminated. Although some third-harmonic components appear, due to the low content of the third-harmonic component, the overall

voltage fluctuation of the SM is still reduced. In addition, due to the addition of a reverse fundamental component, the fundamental frequency fluctuation of the capacitor voltage is reduced. Therefore, when the circulating current has the predicted value, the capacitor voltage fluctuation is smaller than the fluctuation when only the secondary circulating current is eliminated. The odd frequency components of the capacitor voltage in the upper and lower bridge arms have opposite directions. To obtain even harmonic components, it is necessary to use $\vec{v}^x_{com} = (\vec{V}^x_{Cp} + \vec{V}^x_{Cn})/2$ to calculate the common-mode voltage $\vec{v}^x_{com}$. At this point, the common-mode voltage only has a secondary component.

Figure 2 shows the simplified optimization hybrid predictive control framework. Only current control is achieved through the MPC, while the circulating current and capacitor voltage fluctuation control is accomplished by the quasi-PR controller described above. A common-mode voltage outer loop controller and a circulating current inner loop controller are used. The duty cycle obtained from the voltage sequencing algorithm is as follows:

$$\vec{M}^x_p = [S_1, S_2, \ldots S_N]^\top, \quad \vec{M}^x_n = [S_{N+1}, S_{N+2}, \ldots S_{2N}]^\top \tag{14}$$

where $S_i = 0, 1$, $i = 1, 2, \ldots, 2N$. Then, the duty cycle obtained by controlling the predicted current adjusted by the output of the quasi-PR controller is as follows:

$$\vec{M}^{x\prime}_p = \vec{M}^x_p - \Delta \vec{M}^x, \vec{M}^{x\prime}_n = \vec{M}^x_n - \Delta \vec{M}^x \tag{15}$$

Finally, the modified duty cycle is fed into the PWM module to obtain the switching signals. In each control cycle, the simplified optimization hybrid control framework can allocate two different switching signals, and the increased switching frequency is represented by the number of switching actions associated with circulating current and capacitor voltage control. This method uses the PR controller in parallel with the MPC controller, eliminating the weight factor design and sequential optimization structure. Due to the optimized parallel structure, the PR controller does not affect the dynamic performance of the power and current control of the MMC. The PR controller increases the switching action related to the circulating current and capacitor voltage, significantly suppressing the circulating current and capacitor voltage fluctuation while retaining good steady performance. However, this method has higher switching frequency and energy loss compared to classical MPC.

## 4. Proposed Adaptive Hybrid Predictive Control Framework

To solve the challenge with the simplified optimization method, we propose an adaptive MPC method. It involves adjusting the number of cost functions calculated by system states to ensure dynamic performance with low computational burden. To solve the problem that the classical method has a higher switching frequency, an adaptive hybrid predictive control framework is proposed. It involves adjusting the SM insertion number through the PR controller to obtain the adjustment amount $\Delta N^x_{pn}$ for the SM inputs. It adds $\Delta N^x_{pn}$ to the selected optimal candidate combination for current predictive control to obtain the ideal SM insertion combination $N^{x opt}_{pn}$. By controlling the threshold $Z_{i_z} = 0.01 \cdot \vec{i}^{x*}_z[k+1]^2$, the SM insertion adjustment amount $\Delta N^x_{pn}$ is adjusted to reduce the change in switching frequency.

### 4.1. The Proposed Adaptive MPC

This work proposes an adaptive MPC to ensure fast dynamic performance. In this method, the candidate insertion number combinations are automatically adjusted according to system operation modes, achieving good adaptation for both steady and transient states. As shown in Figure 4, at the stage of predictive current control, the system operation modes are distinguished between steady and transient states. If the system is operating in a steady state, three insertion number combinations $(N^x_p[k], N^x_n[k])$, $(N^x_p[k] + 1, N^x_n[k] - 1)$, and $(N^x_p[k] - 1, N^x_n[k] + 1)$ are considered in the cost function of grid current. Once the cost function values of the three candidate combinations are all larger than the threshold

$Z_{i_g} = 0.01 \cdot \vec{i}_g^{x*}[k+1]^2$, it indicates that the system is operating in a transient state. Next, two additional candidate combinations ($N_n^x[k]$, $N_p^x[k]$ and $0.5N$, $0.5N$) are calculated to shorten the optimization steps and improve the dynamic performance. As shown in Figure 3, during the steady state, there are still only three candidate combinations considered in the cost function calculations. However, when the system reference suddenly changes, the proposed method calculates more combinations for the SM inputs, which improves the dynamic performance of the system. In summary, this method has low computational complexity in the steady state and only adds two to four cost function calculations in the transient state, effectively balancing computational complexity and dynamic tracking performance. Algorithm 1 is the proposed system state judgment algorithm.

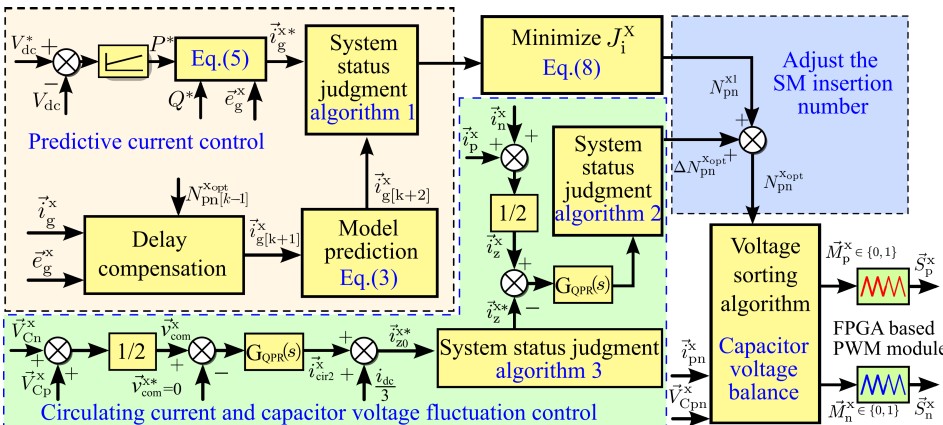

**Figure 4.** The adaptive hybrid predictive control framework.

---

**Algorithm 1** State judgment in phase current control.

1: **function** A1 $((N_p^x[k], N_n^x[k]), (N_p^x[k]+1, N_n^x[k]-1), (N_p^x[k]-1, N_n^x[k]+1), Z_{i_g})$
2: **if** $J_i^x(N_p^x[k], N_n^x[k]) < Z_{i_g}$ && $J_i^x(N_p^x[k]+1, N_n^x[k]-1) < Z_{i_g}$ && $J_i^x(N_p^x[k]-1, N_n^x[k]+1) < Z_{i_g}$ **then**
3:      Calculate $J_i^x(N_n^x[k], N_n^x[k])$, $J_i^x(0.5N, 0.5N)$;
4: **else**
5:      $J_i^x(N_n^x[k], N_n^x[k])$=99999;
6:      $J_i^x(0.5N, 0.5N)$=99999;
7: **end if**
8: **Output:**   $J_i^x(N_p^x[k], N_n^x[k])$,   $J_i^x(N_p^x[k]+1, N_n^x[k]-1)$,   $J_i^x(N_p^x[k]-1, N_n^x[k]+1)$, $J_i^x(N_n^x[k], N_n^x[k])$, $J_i^x(0.5N, 0.5N)$;
9: **end function**

---

### 4.2. Proposed PR Circulating Current and Capacitor Voltage Fluctuation Suppression Controller

The simplified optimization hybrid predictive control framework can effectively suppress circulating current and capacitor voltage fluctuation, but it increases switching frequency and energy loss. To solve the problem, we maintained one switch state per cycle and developed an adaptive hybrid predictive control framework. We added a PR controller that modifies the SM insertion number instead of the duty cycle without increasing the switching frequency. When the system is in a steady state, the range of adjustment $\Delta N_{pn}^x$ for the SMs is small, and when the system is in a transient state, $\Delta N_{pn}^x$ is expanded to suppress circulating current and capacitor voltage fluctuation.

First, the candidate combinations are selected in the predictive current control. The candidate combination ($N_p^{x1}[k+1]$, $N_n^{x1}[k+1]$) is the optimal candidate combination in the grid current control. When the cost function of the candidate combination is higher than the threshold value of $Z_{iz}$, the system is in a transient state, and in the opposite case, it is in a steady state. Figure 4 shows the improved hybrid control framework. Algorithm 2 is the

algorithm for controlling the limit of SM input adjustment. Algorithm 3 is the proposed algorithm for selecting a circulating current reference based on control objectives.

---

**Algorithm 2** State judgment in circulating current control.

---

1: **function** A2 $((N_p^{x1}[k+1], N_n^{x1}[k+1]), Z_{i_z}, \Delta N_{pn}^x)$
2: **if** $J_z^x(N_p^{x1}[k+1], N_n^{x1}[k+1]) < Z_{i_z}$ **then**
3:　　limit $-4 \le \Delta N_{pn}^x$ && $\Delta N_{pn}^x \le 4$;
4: **else**
5:　　limit $-2 \le \Delta N_{pn}^x$ && $\Delta N_{pn}^x \le 2$;
6: **end if**
7: **Output:** $\Delta N_{pn}^x$;
8: **end function**

---

**Algorithm 3** State judgment in capacitor voltage control.

---

1: **function** A3 $(i_{dc}, V_{dc}, N, \vec{V}_{com}^x, \vec{i}_z^x, i_{z0}^{x*})$
2: **if** $V_{com}^x / (V_{dc}/N) \le \vec{i}_z^x / (i_{dc}/3)$ **then**
3:　　let $i_z^{x*} = 0$;
4: **else**
5:　　let $i_z^{x*} = i_{z0}^{x*}$;
6: **end if**
7: **Output:** $i_z^{x*}$;
8: **end function**

---

Step 1 : In the predictive current control, as before, the current tracking is achieved based on Equation (8). Then, the cost function is minimized to select the optimal SM insertion number $N_{pn}^{x1}$ for the current stage.

Step 2: In the PR-based control loop, the circulating current and circulating current reference are calculated and sent to the PR controller.

Step 3: The output of the PR controller is the corrected value $\Delta N_{pn}^x$ for the SM insertion number, which effectively suppresses the circulating current through $\Delta N_{pn}^x$.

Step 4: Finally, the controller adds the SM insertion number $N_{pn}^{x1}$ to the correction value $\Delta N_{pn}^x$ for the SM insertion number, considering the circulating current (the ideal SM insertion number is shown in Equations (14) and (15)). Then, the sorting algorithm balances the capacitor voltage and output switching signals.

$$N_p^{x_{opt}} = N_p^{x1} + \Delta N_{pn}^x, \quad N_n^{x_{opt}} = N_n^{x1} + \Delta N_{pn}^x \tag{16}$$

By taking the derivative of Equations (12) and (13), it can be seen that the amplitude of the circulating current and capacitor voltage is inversely proportional when $\theta_2 = 0$. When $i_z^{x*}$ is constant, $\theta_2 = 0$ and the capacitor voltage is the smallest. When $\theta_2 = 0$ and $i_z^{x*} = i_{dc}/3 + m\vec{i}_g^x \sin(2\omega_g t + \varphi + \pi/2)/4$, the capacitor voltage fluctuation is the smallest. When $\theta_2 = 0$ and $i_z^{x*} = i_{dc}/3 + m\vec{i}_g^x \sin(2\omega_g t + \varphi + \pi/2)/4$, the circulating current is the smallest. Due to the quarter-cycle difference between the peak values of the circulating current and capacitor voltage, the peak values of the circulating current and capacitor voltage fluctuation can be suppressed separately by changing the reference of the circulating current. When the circulating current reaches its peak, it is suppressed, and the capacitor voltage fluctuation is not severe. The reverse is also true.

1: If $\vec{v}_{com}^x / (V_{dc}/N) > \vec{i}_z^x / (i_{dc}/3)$ and $|\vec{v}_{com}^x / (V_{dc}/N)| > 1.05$, it means that the capacitor voltage fluctuation is greater than the circulating current deviation, and the system is in a transient state; excessive deviation in the capacitor voltage fluctuation from the reference requires priority control. Then, we use a $\vec{v}_{com}^x$ outer loop PR controller to obtain $i_z^{x*}$ and $-4 \le \Delta N_{pn}^x \le 4$ to ensure the control quality of the capacitor voltage fluctuation control.

2: If $\vec{v}^{x}_{\text{com}}/(V_{\text{dc}}/N) > \vec{i}^{x}_{z}/(i_{\text{dc}}/3)$ and $|\vec{v}^{x}_{\text{com}}/(V_{\text{dc}}/N)| \leq 1.05$, it means that the capacitor voltage fluctuation is greater than the deviation in the circulating current, and the system is in a steady state; the capacitor voltage fluctuation does not deviate excessively from the reference. Then, we use a $\vec{v}^{x}_{\text{com}}$ outer loop PR controller to obtain $i^{x*}_{z}$ and $-2 \leq \Delta N^{x}_{\text{pn}} \leq 2$ to ensure the control quality of the predictive current control.

3: If $\vec{v}^{x}_{\text{com}}/(V_{\text{dc}}/N) \leq \vec{i}^{x}_{z}/(i_{\text{dc}}/3)$ and $J^{x}_{z} > Z_{i_{g}}$, it means that the circulating current is greater than the capacitor voltage fluctuation deviation, and the system is in a transient state; excessive deviation in the circulating current from the reference requires priority control. We set $i^{x*}_{z} = 0$ and $-4 \leq \Delta N^{x}_{\text{pn}} \leq 4$ to ensure the control quality of the circulating current control.

4: If $\vec{v}^{x}_{\text{com}}/(V_{\text{dc}}/N) \leq \vec{i}^{x}_{z}/(i_{\text{dc}}/3)$ and $J^{x}_{z} \leq Z_{i_{g}}$, it means that the circulating current is greater than the capacitor voltage fluctuation deviation, and the system is in a steady state; the circulating current does not deviate excessively from the reference. We set $i^{x*}_{z} = 0$ and $-2 \leq \Delta N^{x}_{\text{pn}} \leq 2$ to ensure the control quality of the predictive current control.

The proposed adaptive hybrid predictive control framework maintains the same combination of switch outputs during each control cycle and has a lower switching frequency than the hybrid MPC with a PR controller. Due to the difference of one quarter of a cycle between the peak value of the circulating current and capacitor voltage, controlling the current reference and coordinating the current and capacitor voltage according to operating conditions can reduce the peak of the circulating current and capacitor voltage. Additionally, this method has better circulating current and capacitor voltage suppression abilities. This is because the output of the PR control is the correction value for the SM insertion number rather than the duty cycle. When the circulating current deviation is severe, the range of action of the PR controller is increased to enhance the circulating current and capacitor voltage fluctuation control performance. Conversely, the range of the PR controller is reduced to enhance the current control performance.

## 5. Simulation and Experimental Verification

We used a three-phase MMC simulation model with $N = 10$ and a grid-connected MMC test bench (shown in Figure 5) with $N = 4$ to test the control performance of the proposed adaptive hybrid control framework. The simulation used DC and AC voltage parameters from the Dogger Bank wind farm in the UK [30]. The difference between the test bench and the topology shown in Figure 1 was that a programmable three-phase AC power supply was used instead of the AC grid, and the DC link of the high-voltage DC system was connected to the resistive load. The system parameters are shown in Table 1.

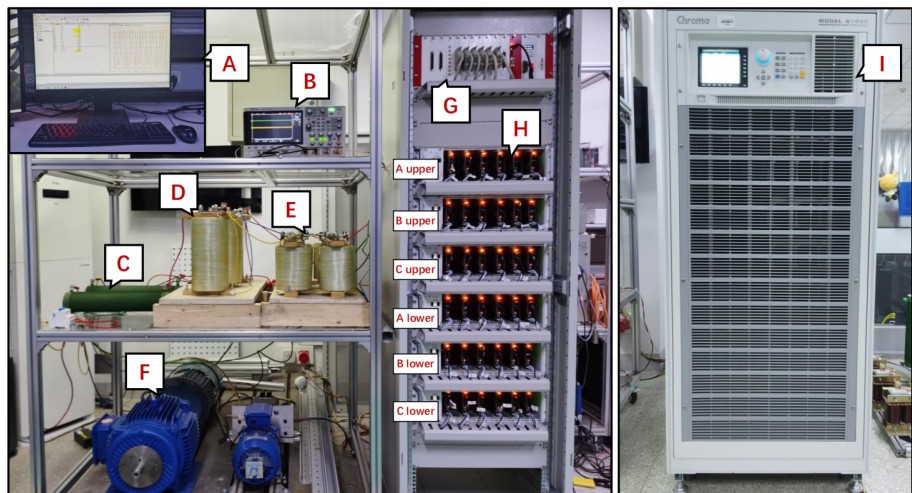

**Figure 5.** Experiment test bench.

**Table 1.** System configuration.

| Parameter | Simulation Value | Experiment Value |
|---|---|---|
| DC-link voltage | $300 \times 10^3$ V | 100 V |
| Number of SMs in arm | 10 | 4 |
| SM capacitor | 0.5 mF | 1.64 mF |
| Arm inductance | 5 mH | 5 mH |
| Grid resistance | 0.5 Ω | 5 Ω |
| Grid inductance | 10 mH | 10 mH |
| Sample time | 100 μs | 100 μs |
| Grid voltage | $60 \times 10^3$ V | 40 V |
| Grid frequency | 50 Hz | 50 Hz |

### 5.1. Overall Validation of the Proposed Method

This section presents the overall control performance of the proposed adaptive hybrid predictive control framework. The test scenario was as follows. The DC resistive load was maintained at 10 Ω. The DC-link voltage was changed from 100 V to 70 V and back to 100 V. The reactive power was changed from 0 Var to 200 Var in 1 s and back to 0 Var in 4 s. Then, it was changed from 0 Var to −200 Var in 5 s and back to 0 Var in 8 s. The overall experimental results are shown in Figure 6, including the DC-link voltage, grid current, output power, bridge arm current and circulating current, bridge arm capacitor voltage, common mode voltage, converter AC voltage, and switching frequency. The proposed method achieved global stability and good steady and dynamic performance. In detail, the proposed method achieved fast tracking of power, the DC-link voltage was stable in both steady- and dynamic-state conditions, and the AC component of the circulating current and capacitor voltage fluctuation was effectively controlled.

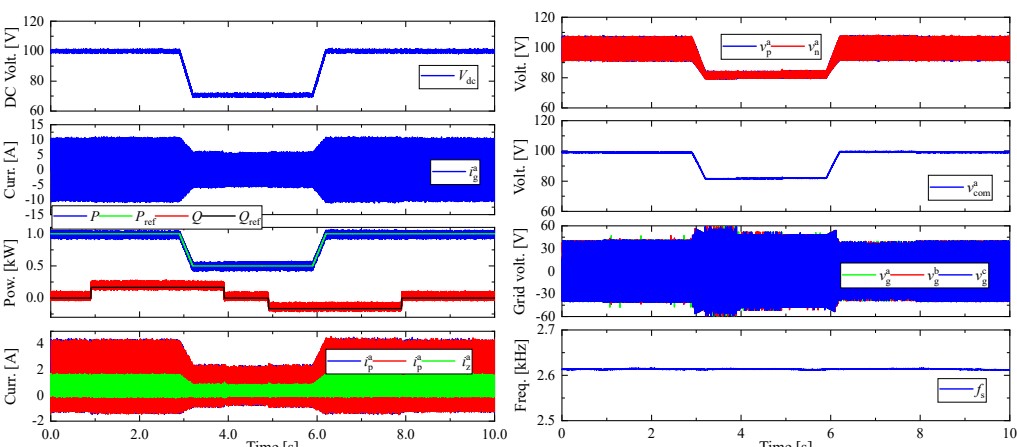

**Figure 6.** Experimental results: the overall control performance with the proposed method.

The improvements achieved with the proposed adaptive MPC were more significant when the SM number $N > 7$; the experimental system was constrained by financial cost and we used $N = 4$. Figure 7d shows the relationship between the time required for active power to follow the reference from zero to rated power and the number of SMs $N$ in each bridge arm. It can be seen that, when $N > 7$, the adaptive MPC has significant differences from traditional methods. It was difficult to demonstrate the dynamic performance difference between the adaptive MPC and the simplified optimization MPC with an $N = 4$ test bench. Therefore, experiments focused on the steady-state performance, circulating current, and capacitor voltage fluctuation suppression were conducted with an $N = 4$ MMC test bench. The $N = 4$ MMC test bench parameters were used for the simulation and to compare the experimental results. An $N = 10$ MMC simulation platform was used to compare the dynamic performance of the traditional and proposed method.

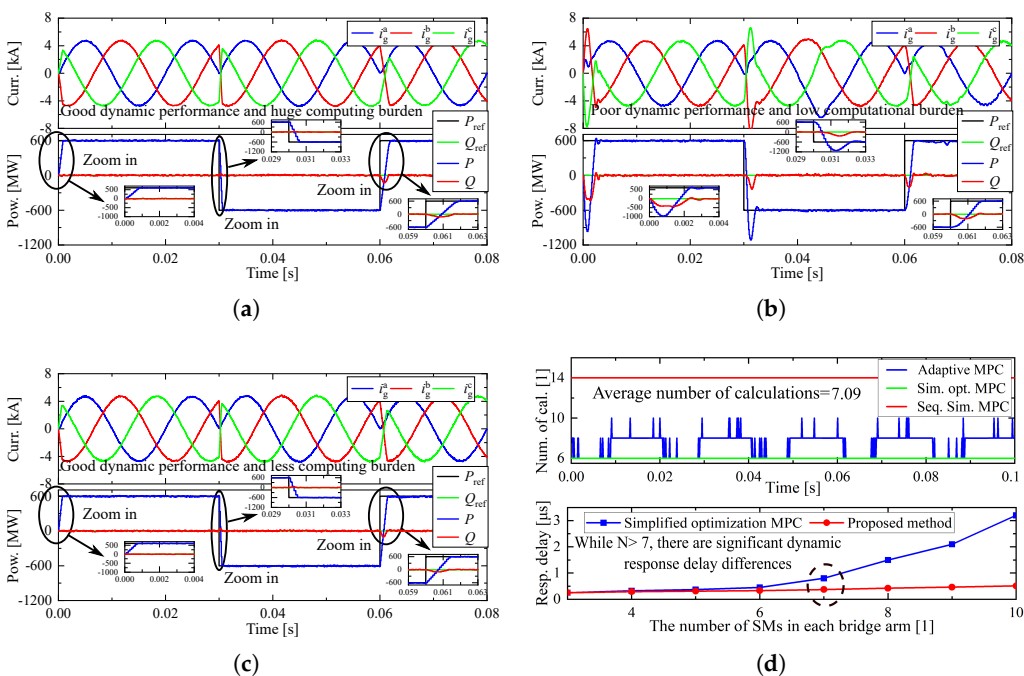

**Figure 7.** Simulation results for the dynamic state control performance: (**a**) sequential optimization MPC; (**b**) simplified optimization MPC; (**c**) the proposed adaptive MPC; (**d**) the relationship between the dynamic response delay, the number of SMs $N$, and the number of cost functions calculated.

*5.2. Dynamic State Control Performance*

This section compares the dynamic performance of the sequential optimization MPC that traverses all combinations [12], the simplified optimization MPC, and the proposed adaptive MPC. To compare the computational burden of the full processes of the two methods, the PR controller was not used. The active power decreased from 600 MW to $-600$ MW in 0.3 s and back to 600 MW in 0.6 s. The grid current and output power are shown in Figure 7a–c. Figure 7d shows the response time and the number of candidate SM input combinations calculated with the three methods.

It can be seen that, during the startup and transient stages, the control performance of the simplified optimization MPC was poor due to the limitation of SM combinations to six. The grid current and output power showed reverse jumps during the startup phase. When the power reference suddenly changed, its power response was slower and overshoot occurred. The proposed method had strong dynamic tracking ability, restoring the response time to a level equivalent to the sequential optimization MPC without overshoot. As shown in Table 2, the switching frequencies of the three methods were similar. The proposed method only needed to calculate an average of 7.09 computations per cycle and had a shorter computation time. The superiority of the proposed method increased with the increase in the number of SMs. In a word, the proposed adaptive MPC balances computational burden while possessing a high-speed dynamic response capability.

**Table 2.** Comparison of the simulation performances of the three methods.

| Method | Sequential Optimization MPC | Simplified Optimization MPC | Proposed Adaptive MPC |
|---|---|---|---|
| Number of calculations | 14 | 6 | 6–10, average 7.09 |
| Response time | 1.1 ms | 2.5 ms | 1.3 ms |
| Computing time (<100 μs) | 81 μs | 56 μs | 62 μs |
| Switching frequency | 2593 Hz | 2576 Hz | 2591 Hz |

### 5.3. Steady-State Control Performance

This section compares the steady-state performance of the hybrid circulating current control framework that does not consider capacitor voltage fluctuation [17], the simplified optimization hybrid predictive control framework, and the proposed adaptive hybrid predictive control framework. The DC voltage and reactive power references were 100 V and 0 Var, respectively. The DC resistor was 10 Ω. Figure 8 compares the steady-state performance of the three methods. The grid voltage and current, output power, and grid current spectrum are shown from top to bottom.

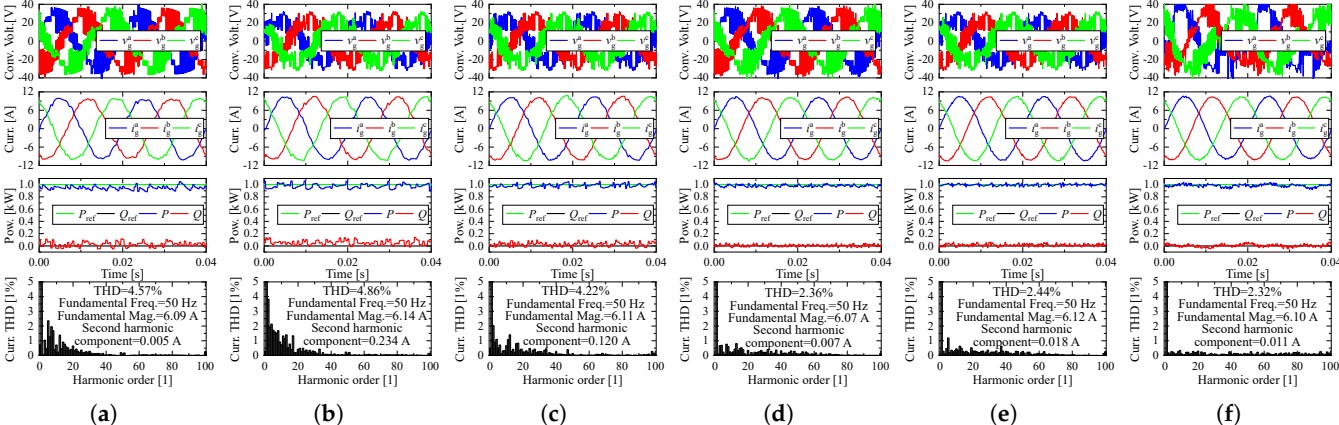

**Figure 8.** The steady-state performance experimental results: (**a**) hybrid circulating current control framework; (**b**) simplified optimization hybrid predictive control framework; (**c**) proposed adaptive hybrid predictive control framework. Simulation results: (**d**) hybrid circulating current control framework; (**e**) simplified optimization hybrid predictive control framework; (**f**) proposed adaptive hybrid predictive control framework.

It can be seen that, compared with the hybrid circulating current control framework, the simplified optimization hybrid control framework could better generated the circulating current with twice the fundamental frequency due to capacitor voltage fluctuation suppression. The circulating current caused a significant twice-frequency harmonic in the grid current, resulting in poor power quality.

The steady-state performance of the proposed method was superior. If the control effects of secondary control objectives, such as circulating current, are poor, the proposed method can increase the output range of the PR controller, improve the adjustment ability for the candidate combinations, and thus enhance the control effect on the circulating current and capacitor voltage fluctuation. Thus, the proposed method has a smaller current THD and better power quality. Additionally, the proposed method controls the circulating current and capacitor voltage in different periods when they reach their peak, which means it can suppress the fluctuation of capacitor voltage while reducing the secondary component of the circulating current and improving power quality. From Table 3, it can be seen that the three methods' calculation times did not exceed 100 μs, and the calculations could be performed with real-time computing on the test bench. The proposed method had a lower computational burden even with the $N = 4$ MMC test bench.

In addition, the proposed method determined the system state during the current control stage and the circulating current control stage, respectively. The candidate combination for the current prediction control expansion and the upper limit of the adjustment for the circulating current control increase were parallel. Thus, the current control did not affect the control performance for the circulating current control, improving the control performance for the subordinate targets, such as circulating current and capacitor voltage.

**Table 3.** Comparison of the experiment performance of the three methods.

| Parameter | Hybrid Circulating Current Control Framework | Simplified Optimization Hybrid Control Framework | Proposed Adaptive Hybrid Control Framework |
|---|---|---|---|
| Peak value of $\vec{i}_z^a$ | 1.19 A | 1.42 A | 1.26 A |
| RMS value of $\vec{i}_z^a$ | 0.67 A | 0.92 A | 0.81 A |
| Peak value of $\vec{V}_{Cn}^a$ | 32.61 V | 31.81 V | 31.32 V |
| Minimum value of $\vec{V}_{Cn}^a$ | 27.45 V | 27.24 V | 27.07 V |
| RMS value of $\vec{V}_{Cn}^a$ | 30.02 V | 29.96 V | 29.39 V |
| Amplitude of $\vec{V}_{Cn}^a$ | 5.16 V | 4.57 V | 4.25 V |
| Peak value of $\vec{v}_{com}^a$ | 30.42 V | 30.04 V | 30.00 V |
| Minimum value of $\vec{v}_{com}^a$ | 29.44 V | 29.89 V | 29.92 V |
| RMS value of $\vec{v}_{com}^a$ | 30.01 V | 29.95 V | 29.95 V |
| Maximum deviation in $\vec{v}_{com}^a$ | 0.57 V | 0.09 V | 0.05 V |
| Computing time (<100 μs) | 57 μs | 52 μs | 53 μs |
| Switching frequency | 2901 Hz | 2896 Hz | 2614 Hz |

*5.4. Comparison of Circulating Current and Capacitor Voltage Fluctuation Performance*

This section compares the circulating current and capacitor voltage fluctuation suppression performance of the hybrid circulating current control framework that does not consider capacitor voltage fluctuation, the simplified optimization hybrid predictive control framework, and the proposed adaptive hybrid predictive control framework. The simulation and experimental results are shown in Figure 9. From top to bottom, the bridge arm and circulating current, capacitor voltage, and circulating current spectrum are shown. Table 3 shows the control performance of the three methods for circulating current and capacitor voltage.

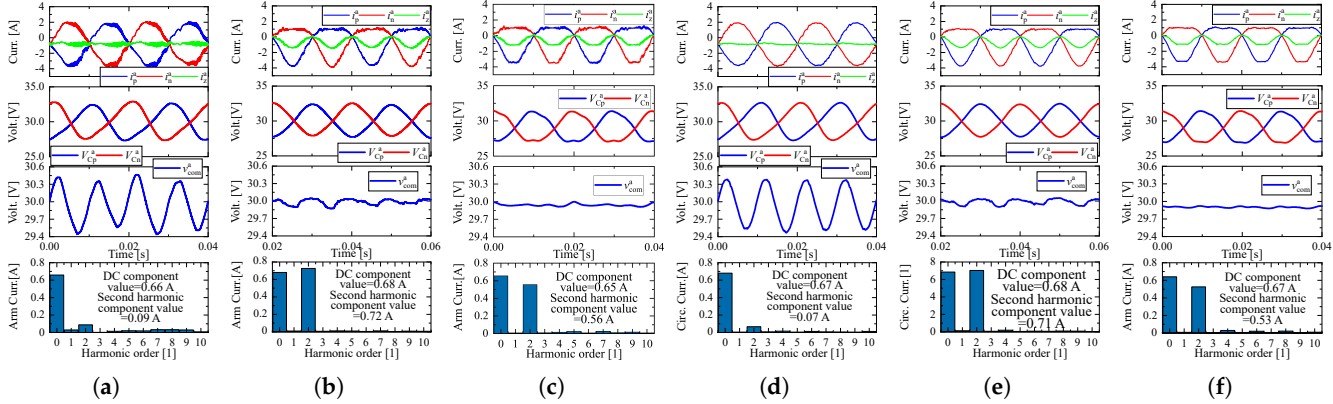

**Figure 9.** Circulating current and capacitor voltage fluctuation suppression performance experimental results: (**a**) hybrid circulating current control framework; (**b**) simplified optimization hybrid predictive control framework; (**c**) proposed adaptive hybrid predictive control framework. [Simulation results:] (**d**) hybrid circulating current control framework; (**e**) simplified optimization hybrid predictive control framework; (**f**) proposed adaptive hybrid predictive control framework.

It can be seen that, although the hybrid circulating current control framework eliminated the circulating current, due to the lack of control over the capacitor voltage, there was an 8.4% fluctuation in the capacitor voltage, and the peak voltage of the capacitor was 8.6% higher than the reference ($V_{dc}/N = 30$ V). There was significant common-mode voltage fluctuation. The need for larger capacitors increases equipment costs.

Due to the capacitor voltage fluctuation suppression in the simplified optimization hybrid control framework, the peak voltage of the capacitor was 6.2% higher than the reference, and the common-mode voltage fluctuation decreased by 84.3%. However, suppressing capacitor voltage fluctuation requires the second-harmonic circulating current. The peak value of the circulating current was relatively high and caused energy losses.

The proposed method can control the circulating current and capacitor voltage separately at different times. When the circulating current is high, the second-harmonic component of the circulating current is suppressed, reducing the RMS value of the circulating current and energy loss. When the capacitor voltage is high, the PR common-mode voltage controller is used to suppress the capacitor voltage fluctuation. Compared with the simplified optimization hybrid control framework, the RMS value for the circulating current AC component with the proposed method was reduced by 33%, and the common-mode voltage fluctuation was reduced by 44%. The peak voltage of the capacitor was only 4.4% higher than the reference. Table 3 shows that, due to the existence of only one switch state per cycle, the proposed method reduced the average switching frequency by 15%, which was close to the result when only the MPC was used. In summary, the proposed method can effectively suppress the capacitor voltage fluctuation and reduce the AC component of the circulating current.

## 6. Conclusions

To address the challenges of optimization with high dynamic performance and low computational burden, balance the capacitor voltage and circulating current, and maintain good performance in circulating current and capacitor voltage control with fewer switching losses, this work proposed an improved adaptive hybrid predictive control framework for MMCs. In the proposed adaptive MPC, the number of candidate combinations is automatically adjusted by distinguishing transient- and steady-state operation modes. The candidate combination number is increased during transient states to ensure good dynamic performance and reduced during steady states to reduce the computational burden. Furthermore, by combining the MPC with linear PR controllers, an improved hybrid control framework was developed, avoiding the sub-optimal control of circulating current and capacitor voltage fluctuation. Due to the correlation between the circulating current and capacitor voltage, this method can achieve coordinated control by switching the circulating current reference of the PR controller. When the deviation in the circulating current is large, it suppresses the circulating current, and when the fluctuation in the capacitor voltage is large, it suppresses the capacitor voltage fluctuation. Noticeably, the output of the PR control is dynamically adjusted based on the deviation in the circulating current and capacitor voltage fluctuation, achieving variable control priority for the grid current and circulating current. In addition, compared with a very recently reported method, this method maintains the single-interval, single-switching characteristics of the MPC and hence has a lower total switching frequency. Finally, experimental and simulation results validated the proposed method's effectiveness.

**Author Contributions:** Conceptualization, Z.Z. and J.L.; methodology, J.L. and Z.L.; software, J.L. and O.B.; validation, J.L. and O.B.; formal analysis, J.L.; investigation, J.L. and Z.L.; resources, J.L.; data curation, J.L.; writing—original draft preparation, J.L. and O.B.; writing—review and editing, J.L. and Z.Z.; visualization, J.L. and Z.L.; supervision, Z.Z.; project administration, Z.Z.; funding acquisition, Z.Z. All authors have read and agreed to the published version of the manuscript.

**Funding:** This research was funded by the National Key R&D Program of China under grant 2022YFB4201700 and in part by the General Program of the National Natural Science Foundation of China under grants 51977124, 52277192, and 52277191.

**Data Availability Statement:** No new data were created or analyzed in this study. Data sharing is not applicable to this article.

**Conflicts of Interest:** The authors declare no conflict of interest.

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
