# Peer review of "Predictive Control of Modular Multilevel Converters: Adaptive Hybrid Framework for Circulating Current and Capacitor Voltage Fluctuation Suppression"

_energies, doi:10.3390/en16155772_

Round 1

Reviewer 1 Report

Predictive Control of Modular Multilevel Converters: Adaptive Hybrid Framework for Circulating Current and Capacitor Voltage Fluctuation Suppression

In this paper, the authors propose an adaptive MPC that adapts the number of candidate combinations to the steady and transient states, significantly reducing the computational burden. Moreover, an improved hybrid combination of MPC with a proportional resonance (PR) controller is used to enhance the circulating current and capacitor voltage fluctuation suppression performance. The idea of this paper is interesting. However, the presentation of this manuscript needs to be improved. Furthermore, I have listed several comments as follows: 

1) The grammar needs to be corrected in a number of places.

2) The state of the art presented in the introduction section needs to be enhanced. Furthermore, the authors need to introduce more recent works. 

3) The authors should test the performance of the proposed method under step change in reactive power reference as presented in the following papers :

[A] Yaramasu, V.; Wu, B.; Chen, J.: Model-predictive control of grid-tied four-level diode-clamped inverters for high-power wind energy conversion systems. IEEE Trans. Power Electron. 29(6), 2861–2873 (2014).

[B] Yaramasu, V.; Wu, B.: Model predictive decoupled active and reactive power control for high-power grid-connected four-level diode-clamped inverters. IEEE Trans. Ind. Electron. 61(7), 3407– 3416 (2014).

[C] Laib, A., Krim, F., Talbi, B., & Sahli, A. (2020). A predictive control scheme for large-scale grid-connected PV system using high-level NPC inverter. Arabian Journal for Science and Engineering, 45(3), 1685-1701. 

4) The output voltage of the multilevel inverter needs to be presented.

5) The computational burden and switching frequency should be measured. 

6) The quality of the figures should be improved. 

1) The grammar needs to be corrected in a number of places.

Author Response

Dear Reviewer
Thanks for your kind comments We strongly agree with your opinions
Due to space limitations and the presence of many images in the response to your opinions. Please see the attachment.

Thank you very much again for your valuable opinions.

Reviewer 2 Report

Multilevel power converters have several significant advantages compared to conventional two-level converters, such as higher energy efficiency, better performance and quality of the generated power, however they require greater complexity and cost in the implementation due to the need for more components and more sophisticated control. Therefore, this work is interesting from the point of view of control optimization.

Some questions are:

1) What power converter configuration is used?. The type of converter can be indicated on paper

2) In Table 1, of simulation parameters it says "DC-link Voltage = 300e3 V" and "Grid voltage 60e3 V", is that correct?

3) A simulation that corresponds exactly to the parameter values used experimentally can be added to compare performance parameters such as THD.

4) The times shown in Table 2 are the computational costs obtained by simulation?. If so, you can also add a measurement of the computational cost experimentally and analyze the minimum and maximum limits of the algorithm's operation times.

5) In Figure 10, where a comparison is made between the proposed method and two others, there you can see that the third harmonic in the proposed method is higher than method "a". Could that have a negative effect?

Author Response

Dear Reviewer
Thanks for your kind comments Due to space limitations and a large number of images in the response to your opinions. Please see the attachment.
Thank you very much again for your valuable opinions.

Round 2

Reviewer 1 Report

I would like to thank the authors for their response to my comments. The revision is generally well done, and the presentation has been improved. In my opinion, this manuscript is suitable for publication in its current form.  

The paper is well-written and well-organized.